# A novel sample handling system for dissolution dynamic nuclear polarization experiments

Thomas Kress[1], Kateryna Che[2], Ludovica M. Epasto[2], Fanny Kozak,[2] Mattia Negroni[2], Gregory L. Olsen[2], Albina Selimovic[2], Dennis Kurzbach[2,*]

[1] Department of Chemistry, University of Cambridge, Lensfield Road, Cambridge CB2 1EW, UK.

[2] University Vienna, Faculty of Chemistry, Institute of Biological Chemistry, Währinger Str. 38, Vienna, Austria.

*Correspondence to: Dennis Kurzbach (dennis.kurzbach@univie.ac.at)*

## Abstract

We present a system for facilitated sample vitrification, melting, and transfer in dissolution dynamic nuclear polarization (DDNP) experiments. In DDNP, a sample is typically hyperpolarized at cryogenic temperatures before dissolution with hot solvent and transfer to a nuclear magnetic resonance (NMR) spectrometer for detection in the liquid state. The resulting signal enhancements can exceed four orders of magnitude. However, the sudden temperature jump from cryogenic temperatures close to 1 K to ambient conditions imposes a particular challenge. It is necessary to rapidly melt the sample to avoid a prohibitively fast decay of hyperpolarization. Here, we demonstrate a sample dissolution method that facilitates the temperature jump by eliminating the need to open the cryostat used to cool the sample. This is achieved by inserting the sample through an airlock in combination with a dedicated dissolution system that is inserted through the same airlock shortly before the melting event. The advantages are threefold: 1. The cryostat can be operated continuously at low temperatures. 2. The melting process is rapid as no pressurization steps of the cryostat are required. 3. Blockages of the dissolution system due to freezing of solvents during melting and transfer are minimized.

*Dedicated to Prof. Geoffrey Bodenhausen on the occasion of his 70th Birthday.*

## 1 Introduction

Dissolution dynamic nuclear polarization (DDNP)(Ardenkjaer-Larsen, Fridlund et al. 2003, Kovtunov, Pokochueva et al. 2018, Jannin, Dumez et al. 2019) is a method used for hyperpolarizing nuclear spins at cryogenic temperatures (Abragam and Goldman 1978) close to 1 K - typically attained in a liquid helium-cooled cryostat at low pressures - coupled to subsequent temperature jump and detection at ambient conditions in a conventional liquid-state nuclear magnetic resonance (NMR) spectrometer. Spin hyperpolarization is herein understood as a strong increase of the population difference between the populations of two eigenstates in a magnetic field $B_0$. The transfer of the hyperpolarized sample from DNP conditions at low temperatures to NMR conditions at ambient temperature is typically achieved with a burst of hot solvent. It rapidly dissolves the sample and pushes it through a capillary to the detection spectrometer. One can thus achieve signal enhancements in liquid state NMR of four orders of magnitude. (Vuichoud, Milani et al. 2015) Capitalizing on the resulting improved sensitivity, DDNP has found various applications in recent years, including real-time metabolomics (Liu and Hilty 2018, Sadet,

Emmanuelle M. M. Weber et al. 2018), reaction monitoring (Boeg, Duus et al. 2019), structural biology (Szekely, Olsen et al. 2018, Wang and Hilty 2019), and detection of long-lived spin states (Tayler, Marco-Rius et al. 2012, Bornet, Ji et al. 2014). However, DDNP instrumentation is still actively being developed to improve its cost-efficiency and reliability, and a need for user-friendly DDNP systems persists.

The sample insertion into and dissolution from the cryostat poses a challenge in designing such systems as both events introduce large heat quantities and warm the instrumentation. The heat shock needs to be compensated by the liquid helium bath within the cryostat (Ardenkjaer-Larsen, Bowen et al. 2019) at the expense of prolonged experimental polarization times or polarization losses.

Indeed, upon insertion of a sample the variable temperature insert (VTI) is typically heated as the sample is warmer than the helium bath and then needs to be cooled down again before efficient DNP can take place. This process can significantly delay the DNP procedure if the VTI is heated too much. Upon dissolution, the VTI often needs to be pressurized so that the dissolution system can be inserted, if a 'fluid-path' system is not available. During this period, the sample also warms up, which might also cause loss of hyperpolarization before the dissolution event due to faster longitudinal relaxation.

In addition, if the temperature of the capillaries used for transfer drops excessively after insertion, the liquid used to dissolve the sample may freeze before exiting the cryostat, preventing the liquid containing the hyperpolarized substance from reaching the NMR spectrometer for detection.

Two widely-used solutions to these problems have been proposed:

1. To minimize the heat-load, Ardenkjaer-Larsen and co-workers have developed a sample insertion and dissolution system based on a 'fluid path' and a 'dynamic seal' as proposed in their original design for the SpinLab DDNP system. (Malinowski, Lipso et al. 2016) The capillaries guiding the dissolution solvent are slowly inserted into the cryostat together with the sample through an airlock. Thus, one can put the capillaries and sample in place without breaking the cryostat's vacuum. After the DNP procedure, the sample can be dissolved through the already positioned capillaries. However, as these are also held at cryogenic temperatures during the DNP build-up period, the dissolution solvent might freeze if the joints between the sample holder and solvent inlet/outlet are not carefully sealed to avoid liquid helium entering the sample chamber.

2. Alternatively, in a second approach inspired by the original 'HyperSense' apparatus, the cryostat is pressurized with helium gas and opened briefly to insert the sample. Bodenhausen and co-workers successfully adapted this design to recently developed DDNP systems. (Kurzbach, Weber et al. 2016, Baudin, Vuichoud et al. 2018) After completing the DNP procedure, the cryostat is pressurized and opened again to insert the capillaries needed for the dissolution event. This design has the advantage of minimizing the risk of freezing the dissolution solvent as the capillaries are not cooled down during the DNP period. However, this comes at the expense of increased heat exchange and helium losses during sample insertion and dissolution compared to the fluid-path design.

To capitalize on both systems, we have developed an alternative hybrid sample handling design. Here, the sample is inserted through an airlock and a vacuum seal system that enables insertion with minimal heat load, while at the same time, sample dissolution can be performed with warm capillaries and without breaking the cryostat's vacuum. We demonstrate this design's implementation in a cryogen consumption-free DNP system similar to that described by Bodenhausen and co-workers (Baudin et al. 2018).

**Figure 1.** a) The cryogen-consumption-free DDNP system (green magnet) used together with the proposed hybrid sample handling system. The low-field spectrometer (blue magnet) used here for detection is situated in the back. b) Zoom on the airlock atop the DDNP system with the vacuum seal attached. The smaller panel shows the detached seal. c) Scheme of the seal (grey), the sample chamber (blue), and the sample tube (white). An array of washers, a silicon fitting, and O-rings renders the seal vacuum-tight, while the sample stick can be moved vertically. The seal is attached to the airlock atop the DNP system via a vacuum nipple. (Images by K. Che and L. M. Epasto.)

## 2 Results and Discussion

The proposed sample handling device is described in Fig. 1. The polarizer, that operates at 6.7 T and a nominal base temperature down to 1.3 K (1.4 K under microwave irradiation), and the spectrometer used for low-field liquid-state detection are shown in Fig. 1a. The most crucial component of the sample handling system is a vacuum seal that surrounds a hollow carbon fiber rod, which we denote as 'sample tube'. The seal is placed atop an airlock *via* a vacuum nipple (Fig. 1b). The seal itself is closed vacuum-tightly around the sample tube via alternating layers of washers and O-rings pressed together by two metal plates (Fig. 1c). A lateral rubber sealing additionally encloses the seal.

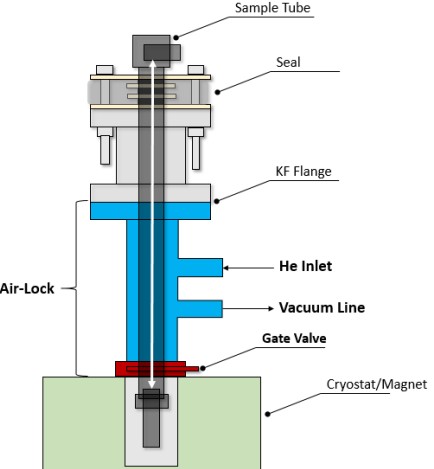

**Figure 2**. Scheme of the airlock system (blue/red) situated between the seal and the cryostat. The latter can be sealed with a gate valve. The sample tube can be inserted upon opening the gate valve. With the He inlet and the connection to a vacuum line, air can be purged from the airlock. The double-headed white arrow indicates the sample tube sliding path through the seal and the airlock into and out of the cryostat (Figure not to scale.)

When mounted on top of the cryostat, the combination of airlock and vacuum-seal allows one to slide the sample tube relative to the seal and position it inside the cryostat without the need to open the latter or break the vacuum within (cf. Fig. 2). A sample can thus be inserted without opening the DNP system to the atmosphere, thereby preventing air from condensing in the cryostat.

The sample tube is closed at its lower end by a 'sample cup' containing the substance to be hyperpolarized, and at its top by a ball valve (Fig. 3a). Hence, it constitutes a closed volume inserted into the cryostat after being flushed with helium gas. The top-end remains at room temperature above the vacuum seal (outside the cryostat), while the sample cup is pushed into the cryostat until it reaches the liquid helium bath where the sample is hyperpolarized.

The leakage rate of our seal been determined to 1.5 +/- 0.5 μL/s at ca. 3 mBar pressure within the probe (the VTI space is sealed from the probe space (Baudin et al. 2018). Generally, the leakage is small enough such that samples can remain in the polarizer for several days without any noticeable air contamination. To avoid ingression of air upon moving the sample tube, it needs to be inserted rather slowly, such that sample insertion takes ca. 5 min (< 5 mm/s). If moved rapidly (*e.g.*, 10 cm/s), the leakage rate rises to > 20 μL/s. A slow insertion has the further advantage to not heat the VTI excessively.

Once hyperpolarized, the sample can be dissolved by opening the ball valve and inserting a 'dissolution stick' that is connected via a PTFE capillary to a pressure heater that provides the hot solvent (here 5 mL of $D_2O$ at a pressure of 1.5 MPa and a temperature of 513 K) used for dissolution. The inner capillary of the dissolution stick has a quite narrow inner diameter of 0.75 mm, such that rather high pressures are needed to dissolve the sample and push it out of the magnet. In addition, the sample has to 'climb' ca. 2 meters in our laboratory for the transfer to some of the spectrometers used for detection. We empirically determined that 1.5 MPa and a temperature of 513 K are feasible to inject the sample directly into an NMR tube waiting in the spectrometer.

Fig. 3a and b show how the dissolution stick inserts into the sample tube and the cup containing the hyperpolarized substance. The inbound and outbound fluid paths are inserted with the dissolution stick such that both are at ambient temperature during the process. Upon insertion of the dissolution stick, the superheated $D_2O$ is squirted

1    onto the sample via the dissolution stick's inner capillary (Fig. 3b), dissolving it and pushing it out of the cryostat.

2    The dissolved sample is ejected through the lumen between the inner capillary and the outer tube.

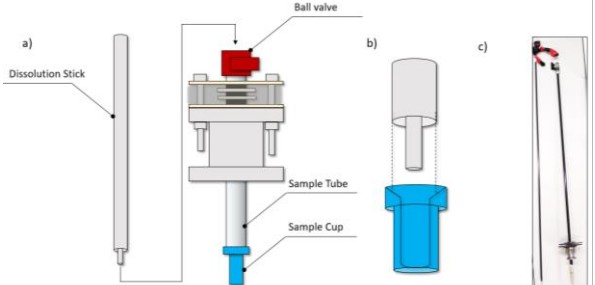

4    **Figure 3.** a) Scheme of the dissolution system (airlock and polarizer omitted for clarity). For a dissolution, the ball valve (red) on top of the

5    sample tube is opened. The sample cup is lifted 100 mm above the liquid helium bath at the bottom of the cryostat. The dissolution stick is

6    then inserted. b) Sketch of how the dissolution stick consisting of two coaxial capillaries (grey) is inserted into the sample cup (blue). The hot

7    solvent is squirted onto the sample through the inner capillary—the lumen between the inner and outer capillary forms the liquid-outlet. c)

8    Image of the sample stick (left) and sample tube (right) with the seal and sample cup attached. (Image by L. M. Epasto.)

9    Fig. 4 displays the path the solvent takes upon dissolution. After melting of the hyperpolarized sample, pressurized

10   helium gas propels the hyperpolarized liquid from the outlet at the top end of the dissolution stick to an NMR tube

11   waiting in a spectrometer for detection. The capillary connecting the DNP and detection spectrometers is

12   surrounded by a copper solenoid that provides a 37 mT magnetic field, as originally devised by Meier and co-

13   workers in the context of so-called 'bullet DNP' (Kouřil, Kouřilová et al. 2018). The solenoid effectively shields

14   the transfer path from low magnetic fields and zero field-crossings in our laboratory that can prohibitively

15   accelerate the relaxation of hyperpolarization. Similar approaches based on 'magnetic tunnels' using permanent

16   magnets have also been successful. (Milani, Vuichoud et al. 2015) These were used here only to cover longer

17   distances to other detection spectrometers (see the Supporting Information). Upon completing the experiment, the

18   sample tube and dissolution stick are removed from the cryostat by sliding both upward through the vacuum seal

19   until the airlock can be closed.

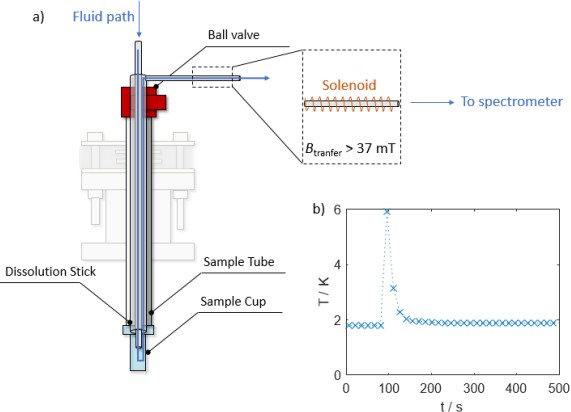

20

21   **Figure 4**. a) Scheme of sample tube with dissolution stick inserted. The inlet for the hot $D_2O$ and the path the fluid takes is marked with a blue

22   arrow. The outlet capillary exiting towards the detection spectrometer is surrounded by a solenoid to maintain a constant magnetic field during

23   the sample transfer. Note that the magnetic field within the solenoid is perpendicular to the field of the DNP apparatus and the detection

24   spectrometer. b) Temperature profile during a dissolution experiment. During the dissolution, the temperature rises by 4 K and subsequently

25   returns to stand-by temperature within a minute.

Fig. 4b displays the temperature changes observed within the cryostat during the dissolution event and demonstrates the relatively low heat-load. The heating is mainly a result of the insertion of the dissolution stick and the dissolution with hot solvent. The sample tube was removed relatively fast (ca. 10 s). It should be noted that our polarizer is smaller than other cryogen-free systems ($\emptyset$40 mm VTI bore size, $\emptyset$12 mm sample space), and so is its capacity to compensate the heat-shock upon dissolution. As a result, the temperature jump can be higher despite a smaller heat load. Further temperature profiles for sample insertions and dissolutions can be found in the Supporting Information. For sample insertions, the heat-load depends on the rate at which the sample is inserted. If inserted slowly (3-5 min) the temperature jump is quite small (typically $< 0.5$ K). If inserted rapidly ($< 1$ min) the VTI temperature can rise by more than 10 K.

The recovered volume after a dissolution experiment is typically up to 4.5 mL out of $5.05 - 5.15$ mL total volume (50-150 $\square$L sample volume + 5 mL hot solvent for dissolution) in our experiments. Ca. 500 $\square$L remain in the capillary system and need to be flushed before the subsequent dissolution. However, only the 600 $\square$L of hyperpolarized solution needed to fill a 5 mm NMR tubes were injected for detection.

Fig. 5 displays hyperpolarized HDO spectra obtained with the proposed system using a sample containing 40 mM TEMPOL in a mixture of 50% glycerol-$d_8$, 40% $D_2O$, and 10% $H_2O$. In this example, a series of 1D NMR signals was detected at one-second intervals on a benchtop spectrometer operating at $B_0 = 1$ T, using a 10° flip angle pulse. The resulting [1]H signal enhancement was $\varepsilon \approx 36\ 000$, corresponding to a polarization of $P\ (^1H) \approx 12\%$. In the solid-state, a polarization of $P\ (^1H) = 15 \pm 3\%$ was achieved at 1.8 K, indicating that ca. 20% of the proton hyperpolarization was lost during the transfer. In contrast, when the solenoid was removed, $P\ (^1H)$ of only ca. 7% was observed, corresponding to a significantly larger ca. 53% polarization loss. Fig. 5a shows how the signal intensity decays after injection into the benchtop NMR spectrometer. Fig. 5b shows the first detected signal immediately after injection overlaid upon the corresponding thermal equilibrium signal detected with the same pulse angle. Fig. 5c shows the decay of the signal enhancement at a 1 s sampling interval. The hyperpolarization decays to naught exponentially with a relaxation rate of $R_1 = 0.21 \pm 0.03$ s$^{-1}$. Polarization levels obtained using the hybrid system presented here are competitive with those previously reported for other dissolution systems. For example, Vuichoud et al.(Vuichoud, Bornet et al. 2016) reported 6% and Ardenkjear-Larsen et al. (Lipso, Bowen et al. 2017) 13% [1]H water polarizations with comparable samples derived from TEMPOL/water-glycerol mixtures.(Leavesley, Wilson et al. 2018) Other recent polarization approaches capable of providing polarization levels of up to 70%, were also reported using samples containing UV-induced radicals. (Pinon, Capozzi et al. 2020)

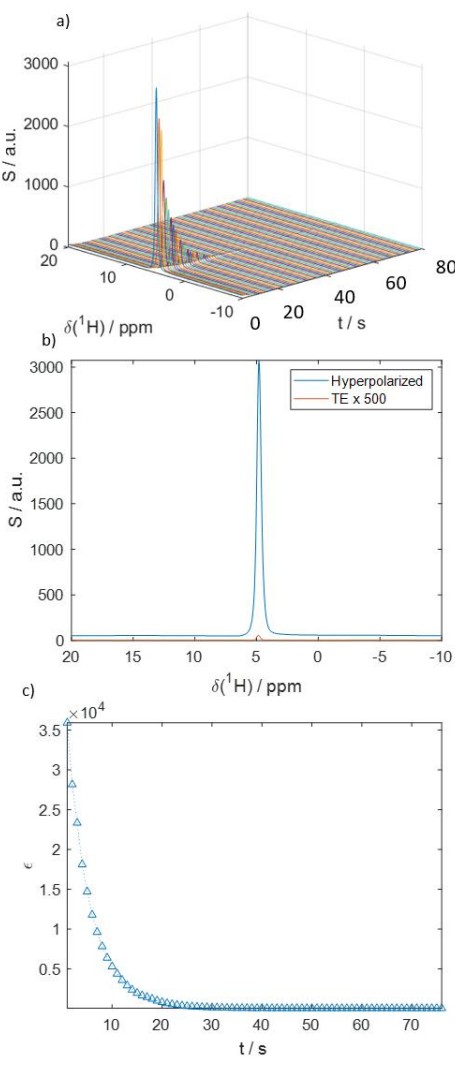

**Figure 5.** a) Time series of $^1$H-detected spectra of hyperpolarized HDO at $B_0 = 1$ T. At t = 0 s the hyperpolarized liquid is injected into the spectrometer. The transfer took ca. 1 s over a length of 1.5 m. b) The HDO spectrum directly after injection (blue) compared to the signal in thermal equilibrium (orange). The signal enhancement was $\varepsilon \approx 36\ 000$, corresponding to a $^1$H polarization of $P\ (^1\text{H}) \approx 12$ %. The detection flip angle was $\alpha = 10°$. c) Decay of the signal enhancement in comparison to thermal equilibrium with time after injection. The mono-exponential decay rate constant $R_1$ was fitted to 0.21 +/- 0.03 s$^{-1}$.

More dissolution-DNP results can be found on the Supporting Information. Data are shown for $^{13}$C-detection of acetate and glycerol-d$_8$, as well as the $^1$H-detection of HDO, on a 500 MHz magnet upon dissolution of a larger, 150 □L sample. For detection with the 500 MHz spectrometer, the samples were transferred through a magnetic tunnel providing a 0.9 T magnetic field, as the samples had to travel longer distances (ca. 4 m). In addition, we show DNP build-up curves for $^{13}$C and $^1$H nuclei at 1.5 K and 3.5 K for TEMPOL concentrations of 40 and 70 mM.

We found that sample transfer problems only occurred as the result of unexperienced operators who failed to couple the dissolution stick with the dissolution cup, leading to leaking of the dissolution solvent into the sample tube. No other modes of failure were observed so far upon dissolution.

The most common 'mode of failure' is the intrusion of air through the seal upon removal of the sample tube after dissolution. If this process is performed too slowly, air can enter the VTI as the bottom end of the sample tube shrinks in diameter during the DNP period and the seal doesn't close tightly anymore around the carbon fiber tube.

**3 Conclusions**

The proposed sample handling system for dissolution DNP has three advantages: 1. The cryostat can be maintained at low temperatures, and the vacuum within is not broken at any stage of the dissolution process. In addition, the heat-load introduced during dissolution is reduced as the dissolution stick does not come into contact with the helium bath. 2. The melting process is very rapid as pressurization of the cryostat is eliminated, in contrast to other HyperSense-inspired systems. 3. Freezing and blockage of the dissolution system are avoided as the dissolution stick is not cooled down at any stage of the experiment. The novelty of our implementation lies in the independent insertion of the dissolution stick while simultaneously maintaining the VTI and the sample space under low pressure. It should be noted that Krajewski et al. (Krajewski, Wespi et al. 2017) also developed a device, that enables to form the contact between sample and dissolution system, while keeping the VTI under low pressures. In their implementation, the layout was designed for multi-sample experiments.

The system is furthermore readily adaptable to different polarizer systems as the vacuum nipple connecting the seal to the DNP apparatus can be adjusted to any flange size. The system is also compatible with narrow sample spaces. For example, the sample tube needs to pass through a bore as narrow as 12 mm in the system presented here.

In conclusion, using this compact and cost-efficient sample handling system, it is possible to perform dissolution DNP experiments with a cryogen consumption-free cryostat without risking quenching of the magnet or introducing air contamination into the cryostat. Moreover, following dissolution, the system reliably returns to its stand-by temperature within a minute, which is a promising step towards higher throughput DDNP.

**4 Experimental**

For DNP 50 □L of a solution of 40 mM TEMPOL in a mixture of 50% glycerol-$d_8$, 40% $D_2O$, and 10% $H_2O$ was hyperpolarized at 1.8 K in a magnetic field of 6.7 T for 2500 s using continuous-wave microwave irradiation at 188.08 GHz. DNP samples were always freshly prepared to avoid ripening effects.(Weber, Sicoli et al. 2018) Fig. 6 displays the build-up kinetics. A VDI microwave source was used together with a 16x frequency multiplier that provided an output power for the microwave of ca. 50 mW. The magnet-cryostat combination was purchased from Cryogenic Ltd. and operated as described in reference (Baudin, Vuichoud et al. 2018).

For detection of the solid-state polarization, a 400 MHz Bruker Avance III system was adapted to a $^1H$ resonance frequency of 285.3 MHz and a $^{13}C$ frequency of 71.72 MHz of by using a broad-band preamplifier for both channels. The detection circuit and the external tune-and-match system were home-built, as described in reference (Baudin, Vuichoud et al. 2018). To monitor the build-up, detection pulses with a flip angle of 1 degree were applied every 5 s.

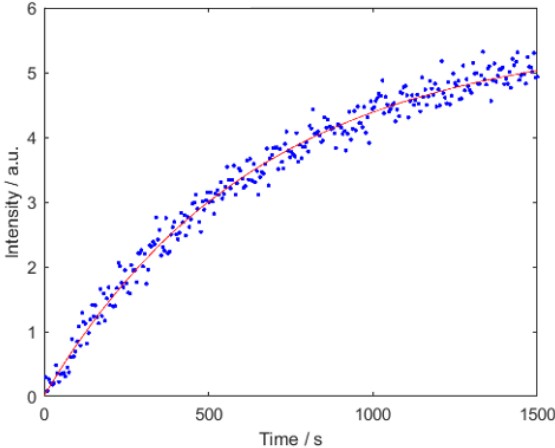

**Figure 6.** Polarization build-up curve at 1.8 K. The build-up rate constant was 0.0016 +/- 0.0001 s$^{-1}$. The polarization was of $P$ ($^1$H) $\approx$ 15 $\pm$- 2%.

After DNP, the sample was dissolved with a burst of 5 mL $D_2O$ at 1.5 MPa as described in the main text. The hyperpolarized liquid was then pushed with helium gas at 0.7 MPa to the detection spectrometer. The dissolution process employed a home-built pressure heater actuated with an Arduino micro-controller. A home-written MATLAB-based user interface controls the dissolution and injection steps.

For low-field detection, the hyperpolarized liquid was transferred to a Magritek SpinSolve Phosphorous spectrometer operating at room temperature and a magnetic field of 1 T. The transfer path was ca. 1.5 m long and the transfer took ca. 1 s. A PTFE capillary with a 1 mm inner diameter and 3.2 mm outer diameter was used. The solenoid (2 turns / mm) surrounding the transfer path provided a >37 mT magnetic field during the sample transfer at a current of 3 A and a power of 450 W. The solenoid ended ca. 500 mm before reaching the bore of the magnet. To avoid heating of the PTFE capillary path, the solenoid was only switched on during the transfer.

A volume of 600 □L of the hyperpolarized liquid was directly injected into an NMR tube that was treated with strongly oxidizing rinsing solutions (Helmanex III) beforehand to reduce the likelihood of gas inclusions forming upon injection into the NMR tube and disrupting NMR detection.(Dey, Charrier et al. 2020)

In the liquid state, $^1$H single pulse acquisitions were repeated at one-second intervals, using a flip angle of 10 degrees. The spectral width was 30 ppm at a carrier frequency centered at 5 ppm. The spectrometer's external lock system was used for referencing the chemical shift.

For high-field detection, the hyperpolarized sample was transferred to a Bruker NEO 500 MHz NMR spectrometer equipped with a Prodigy BBFO probe. Again, a volume of 600 □L of the hyperpolarized liquid was directly injected into an NMR tube that was treated with strongly oxidizing rinsing solutions (Helmanex III) beforehand. Pulses with 1° flip angles for $^1$H detection and 5° flip angles for $^{13}$C detection were applied every second for detection.

All data were processed with home-written scripts using the MATLAB 2019 software package or Bruker TopSpin 4.0. All data were zero-filled and apodized with a Gaussian window function before Fourier transformation.

**Acknowledgments**

The acknowledge support by the NMR core facility of the Faculty of Chemistry, University of Vienna, and thank Profs. Sami Jannin and Benno Meier for helpful discussion. The project leading to this application received funding from the European Research Council (ERC) under the European Union's Horizon 2020 research and innovation programme (grant agreement 801936). This project was supported by the Austrian FWF (stand-alone grant no. P-33338).

**Author contribution**

TK, KC, LME, FK, MN, GO, AS and DK built the DDNP system and performed experiments. DK wrote the manuscript with the help of all authors.

**Competing interest**

The authors declare no conflict of interest.

**Data availability**

All data are available under DOI: 10.5281/zenodo.4738932.

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
