# Peer review of "A novel sample handling system for dissolution dynamic"

_Magnetic Resonance, 2021_

## Author Response (AR1)

To
Prof. Gottfried Otting
Magnetic Resonance

[Figure]

Assoc. Prof. Dr. Dennis Kurzbach
[www.Vienna-DNP.at](www.Vienna-DNP.at)
Universität Wien
Fakultät für Chemie
Institut für biologische Chemie
Währinger Straße 38
1090 Wien
Mail.: Dennis.Kurzbach@univie.ac.at
Tel.: +43-1-4277-70528

Vienna, the 07.05.2021

**Response to referee comments for manuscript mr-2021-12**

Dear Prof. Otting,

Please find enclosed a revised version of the manuscript by Sicoli et al. entitled

**A novel sample handling system for dissolution dynamic nuclear polarization experiments**

That we would like to resubmit for publication in the Magnetic Resonance. We have responded to all questions and concerns raised by the referees (see below, referee comments in black, our responses in blue). Additionally, we have uploaded all data to a FAIR server and updated all references.

We hope that you find our work now suitable for publication in your journal.

Sincerely,

Dennis Kurzbach

**Referee 1**

The manuscript by Kress et al. entitled "A novel sample handling system for dissolution dynamic nuclear polarization experiments" entails a discussion of a new method of sample retrieving and handling technique in dissolution dynamic nuclear polarization (DNP) in which a three-fold advantages were noted over currently used dissolution method: (1) cryostat operation is uninterrupted, (2) dissolution does not require overpressurization of the sample space, (3) the use of a confined airlock minimizes freezing and blockages in case of dissolution mishaps.

Based from my experience in homebuilt dissolution DNP instrumentation assembly, dissolution mishaps (e.g. hot solvent leak into the cryostat) can result to freezing of dissolution stick, high boiloff of liquid Helium bath, and in some cases, calling the day off for experiments since the dissoltuion stick is stucked in the cryostat. This can also be problematic even for commercial systems such as hypersense in which accidental spray of superheated water or solvent into the cryostat, often requiring a visit by technical engineer for repair. The confined airlock technique presented here by the authors is a novel way to bypass these potential issues and it sounds like the low vacuum pressure in the cryostat sample space is maintained during dissolution which implies that the next DNP sample can be brought to low temperature rather relatively quickly. In light of this new development that solves current challenges in dissolution DNP, I highly recommend publication of this manuscript by Kress et al., with minor revision and suggestions:

Page 1 abstract, line 10: change "For DDNP," to "In DDNP,"

Page 1 abstract, line 15: change "Here," to "Herein,"

Page 1, line 28: change "here" to "herein"

Page 1, line 46: change "widely used" to "widely-used"

Page 2, line 1: put a comma after "To minimize the heat load"

Page 2, line 27: Change the first 3 words to "Herein, we demonstrate"

Page 5, Figure 5, lines 2: indicate the unit "t = 0 s".

We will correct this.

In addition, I have a question for the authors: is there any particular reason why the hyperpolarization was done the 1H rather than the staple 13C tracers in DDNP? This manuscript is self-sufficient and great in its current form with the 1H studies--thereby recommended for publication, but I was just wondering why 1H was measured instead of 13C spins in which majority of the DDNP metabolic imaging groups are working on.

We presented $^1$H data, since the low-field spectrometer used for detection cannot detect $^{13}$C nuclei, and hyperpolarized water can provide a valuable help in protein NMR to study folded sites (Szekely et al. J. Am. Chem. Soc. 2020, 142, 9267−9284). However, we have also recorded $^{13}$C data on another spectrometer ($^{13}$C-labeled acetate and natural-abundance glycerol data detected at 11.8 T), which will be added to the manuscript.

**Referee 2**

The authors present a potentially very interesting work where they study a specific configuration for dissolution DNP. Hyperpolarization and DDNP has attracted much attention and is an important subject in magnetic resonance. However, the authors present only spare data and an insufficient description of their system. The technical description should be further detailed. Data should be presented that not only support the claim of robustness, but thoroughly characterizes the performance of the system (polarization for more samples and conditions, sample recovery, volume, temperature, …).

The data requested by the referee has been recorded. As the referee asks for temperature and volume variations, we assume that he refers to low-temperature conditions, although unrelated to the presented sample handling system. We have recorded the following solid-state data at various radical concentrations, temperatures and sample volumes:

$^1$H and $^{13}$C polarization build-up at 1.4, and 3.5 K (40 mM and 70 mM TEMPOL concentrations), sample volumes of 50-150 uL, dissolution and injection into a 1 T as well as 11.8 T spectrometer.

For all samples, temperature profiles upon insertions and dissolutions are available.

We assume that 'recovery volume' refers to the volume ejected from the polarizer upon dissolution. This is typically up to 4.5 mL out of 5.05 – 5.15 mL total volume (50-150 uL sample volume + 5 mL hot solvent for dissolution) in our experiments. Ca. 500 uL remain in the capillary system. However, only 600 uL sample are injected in the 5 mm NMR tubes.

P.1 L.29: The spins are not parallel or antiparallel to the magnetic field. All spins are in a superposition of the two eigenstates. Hyperpolarization is a strong increase of the population difference between the populations of the two eigenstates (or a strong net alignment of the spins with the magnetic field).

We will rephrase this sentence following the referee's definition as 'a strong increase of the population difference between the populations of the two eigenstates.'

P.1 L.41: Please elaborate on the link between the heat shock and the prolonged polarization time and dissolution loss?

Two points need to be considered in this regard: 1. Upon insertion of a sample the variable temperature insert (VTI) is typically heated as the sample is warmer than the helium bath. The VTI then needs to be cooled down again before efficient DNP can take place. This process can delay the DNP procedure if the VTI is heated too much. 2. Upon dissolution the VTI often needs to be pressurized so that the dissolution system can be inserted, if a 'fluid-path' system is not available. During this period, the sample warms up, which might also cause loss of hyperpolarization before the dissolution event due to faster $T_1$ relaxation.

P.2 L.22: The sentence is written as a fact rather than speculation. Is there any evidence in literature or this work that supports better robustness of configuration 2? The authors should state the argument in principle and present data in support of their own work. What is the robustness of their implementation (1 out of 100 failures, 2, 5, 10, …)? What are the failure modes?

We realize that this sentence was misleadingly phrased. We do not wish claim that configuration 2 is any more robust then the fluid path. However, in our hands it was not easy to avoid freezing of the sample upon dissolution, if both capillaries are cooled during the DNP period as in configuration 1. A sophisticated sample holder incl. a disposable Teflon seal as published by Capozzi and Ardenkjaer-Larsen can yet overcome this freezing problem.

Our implementation provides an alternative solution for the freezing problem.

Concerning the robustness of our implementation, up to now, 5 dissolutions failed out of ca. 50 experiments that we performed, as the dissolution stick wasn't held "tightly enough" by an unexperienced operator leading to leaking of the dissolution solvent into the sample tube. All other experiments worked fine. No other modes of failure were observed so far upon dissolution.

A major 'mode of failure' that can be noted though would be air intrusion through the seal upon removal of the sample tube after dissolution. If this process is performed too slowly, air can enter the VTI as the bottom end of the sample tube shrinks in diameter during the DNP period and the seal doesn't close tightly enough anymore around the carbon fiber tube.

P.3 L.10: Can the authors provide any data on the leak rate (static and dynamic) of the described seal (mbar L/s)? Or say something about the ingression of air and its potential effect? How fast can the sample tube move?

The leakage rate has been determined to 1.5 +/- 0.5 µL/s at ca. 3 mBar pressure within the probe (the VTI space is sealed from the probe space as described by Baudin et al. JMR 2018). Generally, the leakage is small enough such that samples can remain in the polarizer for several days without any noticeable air contamination. To avoid ingression of air upon moving the sample tube, it needs to be inserted rather slowly, such that sample insertion takes ca. 5 min (< 5 mm/s). If moved rapidly (e.g. 10 cm/s), the leakage climbs to > 20 µL/s. A slow insertion has the further advantage to not heat the VTI excessively.

P.3 L.19: 1.5 MPa (15 bar) and 513 K (240 C) seems very excessive compared to other polarizer systems? Please comment on the choice of dissolution conditions and what the effect would be of operation at lower temperature and pressure.

The inner capillary of the dissolution stick has quite narrow inner diameter of 0.75 mm, such that higher pressures are needed to dissolve the sample and push it out of the magnet. In addition, the sample has to 'climb' ca. 2 meters in our laboratory for the transfer to some of the spectrometers used for detection. We empirically determined that this pressure and temperature are feasible to inject the sample directly into an NMR tube waiting in the spectrometer.

P.4 L.2: 37 mT is much less than the field generated by magnetic tunnels based on permanent magnets and the field is too low to effectively avoid T1 shortening by the nitroxide radical. Later it is stated that 25% of the polarization is lost during the approx. 1 s transfer time. Please comment on the choice of magnetic field and the effectiveness.

The problem we encountered was that of several zero-field crossings between the DNP magnet and the detection spectrometer. The shortest path between the DNP and detection magnet leads the sample through these crossings. We empirically found that applying a constant field, even if it is only a few mT (e.g., Meier and co-workers used 75 mT, which was sufficient even for solid samples; *Nat Commun* **10,** 1733 (2019), Jannin et al. reported 4 mT:

https://chemrxiv.org/ndownloader/files/27462887), within the solenoid reduced the polarization loss due to zero-field crossings. Besides, we agree with the referee, a magnetic tunnel using permanent magnets is more efficient. We have installed one to cover longer distances with a field of 0.9 T. However, the problem of zero field-crossing remains between the exit of the tunnel and the bore of the magnet when using magnet with the Bruker Ultra-Shield technology for solution-state detection.

P.4 L.14: Fig. 4b would be easier to read if the x-axis was expanded, e.g. 0 to 300 s. The temperature increase seems quite substantial (approx. rising to 6 K). This does not seem to be a small heat load? Is this due to the slow retraction of the sample tube? How fast is it removed?

The heating is mainly a result of the insertion of the dissolution stick and the dissolution with hot solvent. The sample tube was removed relatively fast (ca. 10 s). It should be noted that our polarizer is smaller than other cryogen-free systems, and so is its capacity to compensate the heat-shock upon dissolution. As a result, the temperature jump can be higher despite a smaller heat load.

P.4 L.16: How big is the sample (mg or uL) and what is the largest size that the sample tube can contain? What is the recovery of the sample in the 600 uL in the NMR tube, i.e. how much is lost or undissolved?

The volume of the hyperpolarized sample is between 50 and 150 uL. The recovery volume after dissolution times is ca. 4.5 mL when 5 mL solvent are used for the dissolution. No visible sample amounts remain in the sample cup upon dissolution. However, 500 uL are 'lost' in the capillaries.

The amount of liquid injected in the NMR tube then depends on the time the sample is pushed through the PTFE capillary to the spectrometer. To inject 600 uL a push time of ca. 1 s was used. To hypothetically inject the entire 4.5 mL ca. 6 s would be needed.

P.4 L.20: Why 1.8 K? It has been stated that the polarizer operates at 1.3 K (P.2 L.28) earlier in the paper.

We realize that this was confusingly phrased. The 1.8 K were simply a particularity of the presented example experiment. We have recorded data between 1.4 and 10 K. The 1.3 K are the nominal base temperature provided by the manufacturer. In our hands, the VTI cannot be cooled below 1.4 K in a continuous operation setup under microwave irradiation. Lower temperatures can only be achieved transiently.

P.4 L.26: The relaxation rate constant is stated to be 0.14 s-1. Fig. 5 states 0.21 s-1. What is correct?

The latter is correct. This will be corrected.

P.4 L.31: It has not been demonstrated that the polarizer is capable of polarizing UV generated radicals. These require fast cold loading not to quench. This claim should be removed unless supporting data can be provided.

We will follow the referee's suggestion and remove this sentence.

Fig 1b does not seem to zoom in on the right part of the main photo?

This is indeed misleading as 2 different implementations of the system are shown (with 3.2 and 8 mm diameter seals). We will update the picture to be clearer.

Conclusion: It is unclear what the operating temperature of the system is (1.3 K or 1.8 K)?

See our above comment. We will phrase this clearer.

The heat load during dissolution does not seem insignificant. What is the heat load during sample loading and how fast can this be done?

The heat-load depends on how fast the sample is inserted. If inserted slowly (3-5 min) the temperature jump is quite small ($< 0.5$ K). If inserted rapidly ($< 1$ min) the VTI temperature can rise by more than 10 K.

Maintaining the sample space a low temperature and avoiding to break the vacuum during sample loading and dissolution has already been demonstrated by several other systems.

We referenced these systems in the manuscript. The novelty of our implementation lies in the independent insertion of the dissolution stick while maintaining the VTI AND the sample under low pressure. We will add this sentence to the manuscript to avoid any further confusion regarding this point.

In addition, Krajewski et al., MRM 77:904–910 (2017) will be added (which we missed to cite before) as the authors present another option for sample handling in DDNP.

No data has been presented on the reliability of the system. How many samples have been dissolved without failure?

So far, ca. 90% of the samples have been dissolved without failure. As stated above, the failure rate is small, so far dissolutions failed only when the dissolution stick wasn't held tightly enough by the operator. However, we missed to mention this in the original submission.